# CMANet: Cross-Modality Attention Network for Indoor-Scene Semantic Segmentation

**DOI:** 10.3390/s22218520

**Published:** 2022-11-05

**Authors:** Longze Zhu, Zhizhong Kang, Mei Zhou, Xi Yang, Zhen Wang, Zhen Cao, Chenming Ye

**Affiliations:** 1School of Land Science and Technology, China University of Geosciences, Beijing 100083, China; 2Key Laboratory of Quantitative Remote Sensing Information Technology, Aerospace Information Research Institute, Chinese Academy of Sciences, Beijing 100094, China; 3College of Computer Science and Technology, Zhejiang University of Technology, Hangzhou 310023, China

**Keywords:** semantic segmentation, indoor scene, HHA data, cross-modality aggregation, attention mechanism

## Abstract

Indoor-scene semantic segmentation is of great significance to indoor navigation, high-precision map creation, route planning, etc. However, incorporating RGB and HHA images for indoor-scene semantic segmentation is a promising yet challenging task, due to the diversity of textures and structures and the disparity of multi-modality in physical significance. In this paper, we propose a Cross-Modality Attention Network (CMANet) that facilitates the extraction of both RGB and HHA features and enhances the cross-modality feature integration. CMANet is constructed under the encoder–decoder architecture. The encoder consists of two parallel branches that successively extract the latent modality features from RGB and HHA images, respectively. Particularly, a novel self-attention mechanism-based Cross-Modality Refine Gate (CMRG) is presented, which bridges the two branches. More importantly, the CMRG achieves cross-modality feature fusion and produces certain refined aggregated features; it serves as the most crucial part of CMANet. The decoder is a multi-stage up-sampled backbone that is composed of different residual blocks at each up-sampling stage. Furthermore, bi-directional multi-step propagation and pyramid supervision are applied to assist the leaning process. To evaluate the effectiveness and efficiency of the proposed method, extensive experiments are conducted on NYUDv2 and SUN RGB-D datasets. Experimental results demonstrate that our method outperforms the existing ones for indoor semantic-segmentation tasks.

## 1. Introduction

Semantic segmentation is one of the essential techniques in scene understanding technologies. It aims to categorize each pixel and assists in the identification and segmentation of scene elements. Initially, the segmentation can be achieved by handcrafted features and machine-learning algorithms [1,2,3], among which, deep learning is the trend of current research [4,5,6,7]. For semantic-segmentation tasks, it is fully recognized that indoor scenes exhibit distinct aspects compared with outdoor scenarios. For instance, services relying on indoor-scene semantic segmentation (indoor navigation, intelligent furniture, etc.) are strongly demanded by individuals. Therefore, facing the aforementioned challenges, some works apply notable features of the images to assist the segmentation, e.g., edges information [8,9]. Additionally, the illumination variations, overlaps among objects, and the imbalanced representations of object categories in indoor scenes always make it impossible to distinguish numerous objects using solely RGB images [10].

Adding the depth information to traditional RGB information with low-cost RGB-D sensors is a conventional way to achieve better performance of indoor-scene semantic segmentation. Nevertheless, the depth images contain only range measurement information, which makes them challenging for feature extraction. As a result, it is natural to employ the three-channel HHA images (three channels represent the horizontal disparity, height above ground, and the angle of the pixel’s local surface normal makes with the inferred gravity direction) [11], which are coded from one-channel depth images. The application of HHA images is demonstrated to be more efficient and robust than one-channel depth images [4,12,13]. The comparison of RGB, depth, and HHA images are shown in Figure 1.

In order to improve feature embedding, it is necessary to consider the essence of data modality, i.e., the quality, attribute, or circumstance of each type. The RGB images describe lightness and saturation, which mainly represent appearance information, whereas the HHA images mainly represent geometric information [14]. As a result, RGB and HHA can complement one another, with HHA discriminating instances and contexts that share similar colors and textures [15], while RGB can assist with indistinguishable structures. As illustrated in Figure 2, the cushion on the sofa has a similar texture to the sofa, so it can be distinguished by the HHA images, and the pictures on the wall have a similar structure as the wall, so they can be distinguished by the RGB images. It is evident that the combination of both modality features from RGB and HHA images can effectively enhance the efficiency of feature embedding.

On the basis of existing deep-learning methods for RGB-D semantic segmentation, two open problems are still widely discussed: how to fuse multi-modality (RGB and depth) features adeptly and how to improve the robustness w.r.t. imperfect data. The first problem is caused by the substantial variations between RGB and depth modalities [12], which lead to inappropriate feature fusion and inferior performance. Some works utilize the depth image as an extra channel [4,16,17], whereas some works extract features independently and fuse them via CNN-based architecture [14,18,19]. Despite this, these methods can only correlate RGB and depth information to a limited extent. Additionally, measurement noise, view angle, and occlusion boundary may affect the RGB sensors (e.g., resulting in overexposure) and depth sensors (e.g., resulting in data loss) during the data collection period, causing the second problem. Therefore, some works [20,21] apply pre-processing to RGB and depth images for denoising and filling in lost depth. However, pre-processing is not stable enough for feature extraction.

In this paper, we propose a novel Cross-Modality Attention Network (CMANet) for indoor-scene semantic segmentation. CMANet is designed under the encoder–decoder architecture. The encoder aims to build multi-step interaction between two modalities and extracts multi-level features from RGB and HHA images. Meanwhile, the decoder enhances the efficiency of feature representation and restores the feature maps to the corresponding resolution step-by-step. It is worth mentioning that the cross-modality ability ensures that CMANet can utilize the essence of different modalities effectively and fuse the RGB and HHA features adeptly. To be specific, the encoder has two parallel branches to extract RGB and HHA features. The parallel design scheme minimizes the unfavorable effects of mutual influence between RGB and HHA features. More importantly, to achieve appropriate feature fusion and filter out the noise of images, we propose the Cross-Modality Refine Gate (CMRG) based on the attention mechanism, weighting the crucial features in the first stage and aggregating them in the second stage. Moreover, the CMRG utilizes the features from different modalities, instead of relying on one modality, which increases the model robustness and achieves a great performance. Additionally, the output of the CMRG is propagated in a multi-step bi-directional operation into both branches to enhance the encoding of features. The decoder is an up-sampled ResNet, which gradually restores spatial resolution and integrates the encoding stage information. Additionally, we conduct pyramid supervision to improve the final semantic performance.

The main contributions of this paper are as follows:We propose a novel RGB-HHA semantic-segmentation network: CMANet. On one hand, CMANet can effectively fuse and extract the cross-modality features; on the other hand, it improves the robustness and efficiency of feature embedding;We design the CMRG based on the self-attention mechanism, which not only filters the noise and highlights advantages in the feature maps, but also facilitates to improve the representation and aggregation of cross-modality information;We conduct extensive experiments on challenging NYUDv2 and SUN RGB-D datasets, on which CMANet evidences its robustness and effectiveness on indoor-scene semantic segmentation.

The rest of the paper is organized as follows. Section 2 briefly reviews the related works. Section 3 gives a detailed description of CMANet, which includes the network architecture, the processing modules, the training and optimization strategies, etc. Section 4 provides the experiment settings, evaluation methods, and the results, along with comparing the proposed method with existing methods and analyzing the strengths and limitations of our approach. Section 5 closes the paper with a brief conclusion and future research considerations.

## 2. Related Works

In this section, we briefly review the literature relevant to our work. The attention mechanism part focuses on the utilization of channel attention and spatial attention, and the RGB-D semantic-segmentation part elaborates the existing methods and essential concerns of cross-modality fusion.

### 2.1. Attention Mechanism

The attention mechanism is derived from the behavior of humans, which is to ignore irrelevant information and pay attention to what is essential. This strategy is commonly applied in the deep-learning field and performs well on Natural Language Processing (NLP) [22,23,24] and Computer Vision (CV) [15,25,26] tasks. Considering the attention mechanism in the CV field, we divide it into two categories based on enhancement type: channel attention (what to focus on) and spatial attention (where to focus on).

In deep learning, the feature maps that propagate in different channels usually represent different features [27], such as texture, boundary, and shape. As a result, channel attention assigns different weights to different channels, determining what features are important (what to focus on), and is widely used [28,29,30,31,32,33]. SENet [31] provides the first step to improving the representation ability of feature maps through channel attention by using the Squeeze-and-Excitation (SE) block. For better modeling capability, GSoP-Net [32] generates attention maps not only by utilizing first-order statistics (i.e., the global pooling descriptor) but by extracting high-order statistics as well. SKNet [33] combines different receptive field features with multiple branches to adapt the weights according to the input feature maps. These methods open up a novel aspect to feature extraction where the valuable information from existing channels is emphasized rather than continuously increasing the number of parameters. Thus, we adopt the channel-attention mechanism in the CMRG to refine the cross-modality feature maps so as to reduce the noise and improve feature representation.

Regarding one feature map, the region attribute can be represented by pixel values in different locations. The spatial attention can adeptly select the interrelated and meaningful regions across the entire map, then reinforce them by allocating higher weights (where to focus on), which has significance for image processing [34,35,36,37,38]. AGNet [36] applies Attention Gates (AGs) for medical image segmentation. The AGs restrain irrelevant regions and highlight significant features via implicit learning. PSANet [37] relates all the positions on the feature map to each other via a self-adaptively learned attention mask, thus alleviating the limitations of convolutional filters with small kernel size. Additionally, transformers have proven to be very effective with NLP and CV tasks [22,39]; therefore, the Vision Transformer (ViT) [38] processes images by cropping each image to 16×16 small samples, which is similar to split a sentence into several words. The spatial attention mainly concerns the crucial regions in one feature map, which means it can enhance the relationship among pixels. Inspired by these works, we consider the fact that RGB and HHA have different geometric properties and propose a cross-modality spatial-attention mechanism that can enhance region characteristics.

In order to better refine the feature maps, CBAM [40] combines the channel and spatial-attention modules, which will be discussed later. Motivated by this, we employ the sequential deployment of the channel and spatial modules for cross-modality features. In this way, the acquisition of what is essential comes not only from latent learning, but also from different physical characteristics from different modalities.

### 2.2. RGB-D Semantic Segmentation

For RGB-D semantic segmentation, not only the category labels are assigned to each pixel; the performance is also enhanced via the cross-modality feature fusion. Based on the existing successful deep-learning RGB semantic-segmentation structures [4,5,6,7], several works on RGB-D semantic segmentation are presented and demonstrate their reliability and practicality.

Since the RGB and depth modalities must be fully utilized in RGB-D semantic segmentation, the cross-modality fusion is crucial. The fusion can be achieved via element-wise summation, concatenation, or a combination of both, and adapted by latent learning [4,13,14,18,19,41,42,43]. For learning common and specific parts from cross-modality features, ref. [14] designs the transformation network between the encoder and decoder, and extracts corresponding features via Multiple Kernel Maximum Mean Discrepancy (MK-MMD). RedNet [18] integrates ResNet [44] and encoder–decoder architecture. In addition, it adds skip connections to optimize decoding and applies pyramid supervision to avoid overfitting. Learning from the structure of RefineNet [45], RDF [13] fuses cross-modality features by applying a Multi-Modal feature Fusion (MMF) network, which is composed of several residual convolutional layers and element-wise summation. To avoid noisy and chaotic information affecting the effectiveness of the network, RAFNet [43] employs a three-stem branch encoder to process RGB, depth, and fusion features, respectively. Meanwhile, RAFNet utilizes the channel-attention model for refinement. The RGB-D semantic-segmentation methods mostly inherit the former deep-learning methods used for image processing, some of which insert the attention mechanism to enhance the features adeptly by filtering noisy and chaotic information. However, the relationship between RGB and depth features has received little interest.

Thus, we propose several processing modules for cross-modality features based on residual convolution and attention mechanism. In this structure, the residual convolution contributes to the latent learning and adjusts adaptively for feature maps, while the attention mechanism contributes to information refinement.

## 3. The Cross-Modality Attention Network

An effective cross-modality network aims to attenuate the image noise and combine the benefits of both RGB and HHA features. Regarding this, a novel attention-based mechanism is introduced in our proposed model (CMANet), and it results in the so-called CMRG design. Additionally, bi-directional multi-step propagation and pyramid-supervision training strategies facilitate the performance. In this section, we will present detailed descriptions of the proposed method in terms of the overall framework, the structure of the Cross-Modality Refine Gate (CMRG), the processing modules, the configuration of the encoder and decoder, and the pyramid-supervision strategy.

### 3.1. Network Architecture

In this subsection, we detail the structure of our CMANet, which includes the overall framework, the processing modules for cross-modality features, and the configuration of the encoder and the decoder.

#### 3.1.1. Overall Framework

Influenced by SegNet [5], our proposed structure is based on encoder–decoder architecture. In the encoder part, latent modality features are extracted from RGB and HHA images, which serve as the input of the decoder. Following that, the decoder gradually reconstructs the high-dimensional features to the original spatial resolution and integrates the encoding stage information through skip connections to produce the results of semantic segmentation. In this structure, the encoder extracts the high-level features while the decoder restores the spatial information, which alleviates the problem of chaos in the semantic assignment in pixel-wise classification.

The architecture of CMANet is presented in Figure 3. The encoder has two CNN branches w.r.t. RGB and HHA. Each branch successively extracts latent modality features from RGB and HHA images. Here, ResNet [44] serves as the backbone for both branches.

In order to enhance the extraction and fusion of cross-modality characteristics, we present the Cross-Modality Refine Gate (CMRG), which is designed based on the self-attention mechanism. The CMRG module receives pairs of encoding feature maps from the RGB and HHA branches (e.g., the outputs of RGB-Layer1 and HHA-Layer1) and produces aggregated features. Regarding the cross-modality fusion, there are several CMRG modules, which correspond to distinct encoding stages. As a result, we can extract more valuable features with the availability of an encoder capable of fusing and boosting features via the separate implementation of encoding and fusion with CMRGs.

After encoding, CMRG5 refines the outputs of RGB-Layer5 and HHA-Layer5 to produce the final pair of feature maps. Nonetheless, the final pairings are the outputs of the encoder, which have different representational capabilities because they are derived from separate modalities and, hence, they cannot be joined by element-wise addition. Consequently, we employ the Context module to aggregate and refine the pairings of high-level feature maps from two modalities.

The decoder component is an up-sampled ResNet backbone consisting of five residual blocks, each of which comprises several Up-sampled Residual Units (URUs). In this structure, the decoder recovers the feature maps to the original spatial resolution stage-by-stage via transposed convolution and combines the features from each encoding stage via Agent modules as skip connections.

The low-level features in the decoder have a better resolution with more position and detail information, but the high-level features contain rich semantic and category information. To improve the use of multi-level features and alleviate the gradient vanishing problem, we generate multi-scale semantic maps from five stages of decoding features for pyramid supervision.

#### 3.1.2. Processing Modules

Here, we outline the processing modules that aid in the propagation of features.

**Residual Units** The residual learning can effectively prevent the degradation of the model and resolve the gradient vanishing issue during back-propagation. The structure of residual units is shown in Figure 4. The Residual Convolutional Unit (RCU) and the Chained Residual Pooling (CRP) are utilized in the Agent and Context modules, which are the sub-components of RefineNet [45], whereas the Downsample Residual Unit (DRU) [18] and Upsample Residual Unit (URU) [44] are applied for the encoder and decoder, respectively. It is worth mentioning that, while the displayed Chained Residual Pooling (CRP) has two blocks in Figure 4b, we only utilize one block in our subsequent modules since one is sufficient for refinement.

**Agent Module** The skip connections between the encoder and decoder are utilized to replenish the detail loss caused by downsampling. Hence, the Agent modules provide certain intermediate addition of the multi-stage feature maps from the encoder to the corresponding decoder layers. The structure of the Agent module is illustrated in Figure 5a. After receiving two feature maps from the RGB and HHA branches, the Agent module first utilizes a 1×1 convolution layer to mitigate the explosion of parameters by reducing the dimension number. Then, each feature map goes through the RCU to adapt the element-wise sum fusion. The combined features are fed to a one-block CRP, where a pooling algorithm spreads large activation values while an additional convolution layer is added to learn the significance of the pooled features. Finally, before passing to the decoder, a Convolutional Block Attention Module (CBAM) is employed to filter and enhance the features. Note that, in order to improve the propagation, we couple the CBAM and residual learning via a shortcut connection.

**Context Module** The Context module is applied to fuse the final outputs of the two CNN branches (RGB and HHA). As illustrated in Figure 5b, the Context module has similar components as the Agent module, but has additional RCUs and a 3×3 convolution layer. The first 1×1 convolution layer reduces the dimension from 2048 to 512. Then, two feature maps are fused by element-wise summation and finally output to the decoder after refinement.

#### 3.1.3. Encoder and Decoder Configuration

The encoder and decoder have different types and numbers of the residual unit. The encoder with the backbone of ResNet-50 utilizes the Downsample Residual Unit (DRU) as illustrated in Figure 4c, with the 1×1 convolution layer with a stride of 2. Regarding the decoder, the residual units are applied to upsample the feature maps, as illustrated in Figure 4d, where the 2×2 convolution layer and the second 3×3 convolution layer both have a stride of 1/2. The encoder and decoder configuration is shown in Table 1, Input denotes the number of input feature channels, Output denotes the number of output feature channels, and Units denotes the number of residual units in this layer.

### 3.2. Cross-Modality Refine Gate

For RGB and HHA data, the former mainly record appearance information (e.g., color, texture) that can emphasize the visual boundary, whereas the latter primarily capture shape information (e.g., structure, spatial) that can highlight the geometric boundary. Thus, it is challenging to fully utilize RGB and HHA images via fusion and enhancement of cross-modality features. We propose the Cross-Modality Refine Gate (CMRG) based on the attention mechanism to aggregate features from multiple modalities.

#### 3.2.1. Convolutional Block Attention Module

As discussed in Section 2, the attention mechanism has been extensively used in the CV field for determining where the focus should be placed and for deciding what is valuable. In particular, the Convolutional Block Attention Module (CBAM) combines channel and spatial-attention mechanisms during propagation, an achieves outstanding performance in feature extraction [40]. For further modification of the CBAM, we discuss its overall structure and sub-modules details first.

The structures of the CBAM and its sub-modules are shown in Figure 6. The CBAM refines the input feature maps by sequentially applying one channel-attention module and one spatial-attention module, as illustrated in Figure 6a. Given input feature maps F, the CBAM first infers a 1D channel-attention map $M_{c}$ using the channel-attention module and refines the feature maps via channel-wise multiplication; then, it infers a 2D spatial-attention map $M_{s}$ using the spatial-attention module and performs spatial-wise multiplication on the channel-refined feature maps to yield the output. The CBAM process can be formulated as follows:(1)$$F^{\prime }=M_{c}\left(F\right)⊗F,$$
(2)$$F_{out}=M_{s}\left(F^{\prime }\right)⊗F^{\prime },$$
where $F\in R^{C\times H\times W}$ represents the input feature maps, $F^{\prime }$ represents the channel-refined feature maps, $F_{out}$ represents the final output, and ⊗ denotes element-wise multiplication. In the multiplication procedure, the attention values are broadcast as follows. Channel-attention values are broadcast along the spatial dimension (channel-wise multiplication), while spatial-attention values are broadcast along the channel dimension (spatial-wise multiplication). Figure 6b,c describe the detailed structure of the channel-attention module and the spatial-attention module.

As shown in Figure 6b, the channel-attention module first aggregates the spatial information of input feature F into two descriptors, average-pooled features $F_{avg}^{c}$ and max-pooled features $F_{max}^{c}$, via average-pooling and max-pooling operations, respectively. Then, both descriptors are supplied to a shared network containing one hidden layer of multi-layer perception (MLP). A reduction ratio is set to the shared MLP in order to decrease parameter overhead. After propagation in the shared MLP, the descriptors are fused by element-wise summation and modified by a sigmoid function to produce the channel-attention map $M_{c}$. At last, the channel-refined features $F_{c}$ can be generated by multiplying the attention map and input features. The channel-attention map is computed as follows:(3)$$\begin{array}{cc} M_{c}\left(F\right) & =\sigma (MLP(AvgPool(F))+MLP(MaxPool(F)))\\ \; & =\sigma (W_{1}\left(W_{0}\left(F_{avg}^{c}\right)\right)+W_{1}\left(W_{0}\left(F_{max}^{c}\right)\right)), \end{array}$$
where σ denotes the sigmoid function, $W_{0}\in R^{C/r\times C}$ and $W_{1}\in R^{C\times C/r}$ represent the MLP weights, and *r* is the reduction ratio. As specified, the MLP weights are shared with both inputs. $M_{c}\in R^{C\times 1\times 1}$ represents the channel-attention map, $F_{avg}^{c}\in R^{C\times 1\times 1}$ represents the average-pooled features, and $F_{max}^{c}\in R^{C\times 1\times 1}$ represents the max-pooled features.

The spatial-attention module is presented in Figure 6c. Similar to the aforementioned channel-attention module, the spatial-attention module aggregates channel information of the refined features to maps $F_{avg}^{s}$ and $F_{max}^{s}$ first via average-pooling and max-pooling along the channel axis. The two maps are then merged by concatenation. After that, the concatenated features are convolved by a standard convolution layer to generate the spatial-attention map $M_{s}$. The final refined features are also produced by multiplication. The spatial-attention map is computed as follows:(4)$$\begin{array}{cc} M_{s}\left(F\right) & =\sigma (f^{7\times 7}\left(\left[AvgPool\left(F\right);MaxPool\left(F\right)\right]\right)\\ \; & =\sigma (f^{7\times 7}\left(\left[F_{avg}^{s};F_{max}^{s}\right]\right)). \end{array}$$
where σ denotes the sigmoid function, $f^{7\times 7}$ represents the 7×7 convolutional layer, and [;] refers to the concatenation. $M_{s}\in R^{1\times H\times W}$ represents the spatial-attention map, $F_{avg}^{s}\in R^{1\times H\times W}$, and $F_{max}^{s}\in R^{1\times H\times W}$ represent the pooled maps.

#### 3.2.2. Structure of Cross-Modality Refine Gate

Although CBAM exhibits good performance w.r.t. the extraction and representation of features through the channel- and spatial-attention mechanisms, it is inadequate for the cross-modality features. Inspired by CBAM, where the channel- and spatial-attention modules are sequentially employed (and the attention mechanisms are applied with each modality separately), here, we design a unique attention module, the Cross-Modality Refine Gate (CMRG), which is designed to deal with cross-modality features. The structure of the CMRG is illustrated in Figure 7. Differing from the CBAM, the CMRG takes the multi-modality features as the input, instead of receiving only one feature maps. The self-attention mechanism in the CMRG utilizes the cross-modality information to produce the attention maps, which only rely on one modality in CBAM.

As shown in Figure 7, the CMRG consists of two parts: the channel-attention module, and the spatial-attention module. The input of the CMRG is a pair of feature maps $F_{RGB}$ and $F_{HHA}$, which are derived from the RGB and HHA branches, respectively. Firstly, the CMRG utilizes the channel-attention module to infer two 1D channel-attention maps—$M_{RGB}^{c}$ and $M_{HHA}^{c}$—to refine $F_{RGB}$ and $F_{HHA}$ via channel-wise multiplication. By the sharing of descriptors from each modality and the inference of exclusive attention maps for each modality, the cross-modality channel-attention operation primarily filters out noisy and chaotic information and improves the effective characteristics of the original feature maps from the channel aspect. Then, the CMRG infers two 2D spatial-attention maps—$M_{RGB}^{s}$ and $M_{HHA}^{s}$—to enhance the channel-refined feature maps $F_{RGB}^{cr}$ and $F_{HHA}^{cr}$ via spatial-wise multiplication. The cross-modality spatial attention primarily strengthens the association among pixels in the feature maps in order to put more attention on the areas with similar characteristics, even though some of them are far from each other. Finally, the output $F_{out}$ is generated by adding the spatial-refined feature maps together. This process can be formulated as follows:(5)$$\begin{array}{c} F_{RGB}^{cr}=M_{RGB}^{c}\left(F_{RGB},F_{HHA}\right)⊗F_{RGB},\\ F_{HHA}^{cr}=M_{HHA}^{c}\left(F_{RGB},F_{HHA}\right)⊗F_{HHA}, \end{array}$$
(6)$$\begin{array}{c} F_{RGB}^{sr}=M_{RGB}^{s}\left(F_{RGB}^{cr},F_{HHA}^{cr}\right)⊗F_{RGB}^{cr},\\ F_{HHA}^{sr}=M_{HHA}^{s}\left(F_{RGB}^{cr},F_{HHA}^{cr}\right)⊗F_{HHA}^{cr}, \end{array}$$
(7)$$\begin{array}{c} F_{out}=F_{RGB}^{sr}+F_{HHA}^{sr}, \end{array}$$
where $F_{RGB}\in R^{C\times H\times W}$ and $F_{HHA}\in R^{C\times H\times W}$ represent the input feature maps, $M_{RGB}^{c}\in R^{C\times 1\times 1}$ and $M_{HHA}^{c}\in R^{C\times 1\times 1}$ represent the channel-attention maps, $M_{RGB}^{s}\in R^{1\times H\times W}$ and $M_{HHA}^{s}\in R^{1\times H\times W}$ represent the spatial-attention maps, $F_{RGB}^{cr}\in R^{C\times H\times W}$ and $F_{HHA}^{cr}\in R^{C\times H\times W}$ represent the channel-refined feature maps, $F_{RGB}^{sr}$ and $F_{HHA}^{sr}$ represent the spatial-refined feature maps, and ⊗ denotes element-wise multiplication. In the multiplication procedure, the attention values are broadcast the same as in the implementation in CBAM.

As shown in Figure 7, the channel-attention module first aggregates each feature maps into two 1D descriptors, totaling four via average-pooling and max-pooling, among which $F_{RGB\_avg}^{c}$ and $F_{RGB\_max}^{c}$ are generated from RGB feature maps, whereas $F_{HHA\_avg}^{c}$ and $F_{HHA\_max}^{c}$ are generated from HHA feature maps. Then, the descriptors from various modalities are concatenated to produce two cross-modality channel descriptors: $F_{avg}^{c}$ and $F_{max}^{c}$. Both cross-modality descriptors are fed into two independent MLPs with one hidden layer: $MLP_{RGB}$ and $MLP_{HHA}$. After the shared network is applied to each descriptor, element-wise summations are utilized to generate modality-specific channel-attention maps $M_{RGB}^{c}$ and $M_{HHA}^{c}$. The channel-refined procedure in CMRM can be formulated as follows:(8)$$\begin{array}{cc} F_{avg}^{c} & =[AvgPool\left(F_{RGB}\right);AvgPool\left(F_{HHA}\right)]\\ & =[F_{RGB\_avg}^{c};F_{HHA\_avg}^{c}],\\ F_{max}^{c} & =[MaxPool\left(F_{RGB}\right);MaxPool\left(F_{HHA}\right)]\\ & =[F_{RGB\_max}^{c};F_{HHA\_max}^{c}], \end{array}$$
(9)$$\begin{array}{cc} M_{RGB}^{c}\left(F_{RGB},F_{HHA}\right) & =\sigma (MLP_{RGB}\left(F_{avg}^{c}\right)+MLP_{RGB}\left(F_{max}^{c}\right))\\ M_{HHA}^{c}\left(F_{RGB},F_{HHA}\right) & =\sigma (MLP_{HHA}\left(F_{avg}^{c}\right)+MLP_{HHA}\left(F_{max}^{c}\right)) \end{array}$$
where σ denotes the sigmoid function and [;] denotes the concatenation. It is worth mentioning that, different from the CBAM, both MLP have weights of $W_{0}\in R^{C/r\times 2C}$ and $W_{1}\in R^{C\times 2C/r}$; *r* is the reduction ratio. Furthermore, $F_{RGB\_avg}^{c}\in R^{C\times 1\times 1}$ and $F_{RGB\_max}^{c}\in R^{C\times 1\times 1}$ represent the RGB descriptors, $F_{HHA\_avg}^{c}\in R^{C\times 1\times 1}$ and $F_{HHA\_max}^{c}\in R^{C\times 1\times 1}$ represent the HHA descriptors, and $F_{avg}^{c}\in R^{2C\times 1\times 1}$ and $F_{max}^{c}\in R^{2C\times 1\times 1}$ represent the cross-modality descriptors.

Following the optimization of the features by the channel-attention module, the pair of feature maps—$F_{RGB}^{cr}$ and $F_{HHA}^{cr}$—are fed into the spatial-attention module. Similar to the implementation of channel attention, the spatial-attention module initially aggregates cross-modality features via average-pooling and max-pooling along the channel axis, and infers four 2D maps, among which $F_{RGB\_avg}^{s}$ and $F_{RGB\_max}^{s}$ are generated from refined RGB feature maps, while $F_{HHA\_avg}^{s}$ and $F_{HHA\_max}^{s}$ are generated from refined HHA feature maps. Then, the concatenation is applied to combine all maps. The cross-modality map is then convolved by two independent standard convolution layers with kernel size of 7×7 to generate the modality-specific spatial-attention map $M_{RGB}^{s}$ and $M_{HHA}^{s}$. The spatial-refined procedure in the CMRG can be formulated as follows:(10)$$\begin{array}{cc} F^{s} & =[AvgPool\left(F_{RGB}\right);MaxPool\left(F_{RGB}\right);AvgPool\left(F_{HHA}\right);MaxPool\left(F_{HHA}\right)]\\ & =[F_{RGB\_avg}^{s};F_{RGB\_max}^{s};F_{HHA\_avg}^{s};F_{HHA\_max}^{s}], \end{array}$$
(11)$$\begin{array}{cc} M_{RGB}^{s}\left(F_{RGB}^{cr},F_{HHA}^{cr}\right) & =\sigma (f_{RGB}^{7\times 7}\left(F^{s}\right))\\ M_{HHA}^{c}\left(F_{RGB}^{cr},F_{HHA}^{cr}\right) & =\sigma (f_{HHA}^{7\times 7}\left(F^{s}\right)), \end{array}$$
where σ denotes the sigmoid function, $f^{7\times 7}$ represents the 7×7 convolutional layer, and [;] refers to the concatenation. $F_{RGB\_avg}^{s}\in R^{1\times H\times W}$ and $F_{RGB\_max}^{s}\in R^{1\times H\times W}$ represent the RGB maps, and $F_{HHA\_avg}^{s}\in R^{1\times H\times W}$ and $F_{HHA\_max}^{s}\in R^{1\times H\times W}$ represent the HHA maps.

To refine the features by taking advantages of different physical significance, the CMRG utilizes combined information from RGB and HHA to generate attention maps instead of relying solely on a single modality. Due to the limited sensing capabilities of the depth camera, this strategy improves the sturdiness and effectiveness of the backbone, especially in the HHA branch. To be specific, for the purpose of generating fine attention maps, the descriptors (maps) are derived using both average-pooling and max-pooling, with average-pooling descriptors (maps) representing the global information and max-pooling descriptors (maps) representing prominent information, thereby improving the availability of the attention maps. In addition, the CMRG employs channel- and spatial-attention mechanisms to improve the representation and aggregation of cross-modality information. In this structure, the channel-attention module is primarily responsible for capturing ’what’ is important, whereas the spatial-attention module is primarily responsible for determining ’where’ should be prioritized.

#### 3.2.3. Bi-Directional Multi-Step Propagation

It should be noticed that the CMRG can properly fuse the features from both branches. Moreover, Bi-directional Multi-step Propagation (BMP) is employed to reduce model complexity and improve propagation efficiency.

BMP propagates the refined results to the next layer in the encoder for more accurate and efficient encoding of the RGB and HHA features by minimizing the fusion result to half its original values rather than adding elements directly. The procedure of the BMP can be formulated as follows:(12)$$\begin{array}{cc} {RGB}_{out} & =(REF+{RGB}_{in})/2\\ {HHA}_{out} & =(REF+{HHA}_{in})/2 \end{array}$$
where REF denotes the output of the CMRG.

### 3.3. Pyramid Supervision

The pyramid-supervision training strategy mitigates the gradient vanishing issue by incorporating supervised learning over multiple levels; furthermore, it utilizes the features in different scales to improve the final semantic performance.

As illustrated in Figure 3, four intermediate side outputs are implemented, which are derived from the features of the four up-layers for pyramid supervision in addition to the final output of the decoder. Each output is generated after the corresponding feature maps are convolved by a 1×1 convolution layer. Unlike the final output, which has the original spatial resolution, the four side outputs have a different spatial resolution, with 1/2, 1/4, 1/8, and 1/16 the height and width of the final output. The loss function of pyramid supervision is formulated as follows:(13)$$\begin{array}{c} Loss\left(O_{1},...O_{5}\right)=\sum_{i=0}^{n}Loss\left(O_{i}\right) \end{array}$$
where
(14)$$\begin{array}{c} Loss\left(O_{n}\right)=\frac{1}{N}\sum_{i}-log\left(\frac{exp(s_{i}\left[g_{i}\right])}{\sum_{k}exp\left(s_{i}\left[k\right]\right)}\right) \end{array}$$
where $Loss(O_{n})$ is the loss function for the final output or the side outputs. $g_{i}\in R$ denotes the class index on the pixel *i* of the groundtruth semantic map. $s_{i}\in R^{N_{c}}$ denotes the vector on the pixel *i* of the output score map and $N_{c}$ denotes the class number of the dataset.

## 4. Experiments

In order to verify the effectiveness of our proposed method, we conduct evaluation experiments on RGB-D public datasets NYUDv2 [20] and SUN RGB-D [21]. To evaluate the results, we compare the semantic-segmentation performance of various methods on three metrics: pixel accuracy, mean pixel accuracy, and mean intersection over union [4]. In addition, ablation experiments are performed on NYUDv2 with ResNet-50 [44] as the backbone, and certain analysis and discussion are provided.

### 4.1. Datasets

In this section, we introduce the public datasets that are utilized in our experiment. The two public datasets are:

**NYUDv2** [20] The NYUDv2 dataset consists of 1449 indoor RGB-D images with dense pixel-wise annotation. According to the official instructions [20], we split them into 795 training images and 654 testing images. A 40-category setting is adopted as in [46].

**SUN RGB-D** [21] The SUN RGB-D dataset contains 10,335 indoor RGB-D images with 37 categories, which include images from NYUDv2 [20], Berkeley B3DO [47], SUN3D [48], and newly captured RGB-D images. We divide the dataset into 5285 training images and 5050 testing images according to the official setting.

### 4.2. Implementation Details

We implement our experiments with Python 3.8 in the Ubuntu 18 operating system with Pytorch [49] framework. All models are trained on one Nvidia RTX A5000 graphics card with batch size of 7. Additionally, we use ResNet-50 pre-trained on ImageNet [50] as the backbone of the two branches in the encoder. We adopt the SGD optimizer with momentum 0.9 and weight decay 0.0005. The initial learning rate is 0.001 and it decays by a factor of 0.8 every 100 epochs. We employ the warm-up strategy in the first 15 epochs. The network is trained for 800 epochs with the NYUDv2 and SUN RGB-D datasets. We set the reduction ratio to 8 in all attention modules.

As for the data augmentation strategy, we utilize random rotation (with 90 or 180 degree clockwise rotation), random flipping (left-right or top-bottom flip), random scaling (with [1.0, 1.6] scale), random color jittering (brightness, saturation, contrast change), and random cropping (to 480×640 pixels).

### 4.3. Experimental Results and Comparisons

We compare our CMANet with existing semantic-segmentation methods on the NYUDv2 and SUN RGB-D datasets. The results of the three aforementioned metrics on the two datasets are displayed in Table 2 and Table 3, whereas the details of class IoU on NYUDv2 are displayed in Table 4.

As shown Table 2, our CMANet outperforms most of the state-of-the-art methods in semantic segmentation on the NYUDv2 dataset. On the three evaluation metrics, CMANet achieves 74.2% pixel accuracy, 60.6% mean accuracy, and 47.6% mean IoU. Additionally, on the most important metric—mean IoU—CMANet achieves a 2.7% improvement compared to RefineNet-101 [45], 1.7% compared to LSD-GF [51], and 0.1% compared to RAFNet [43]. It is noticed that we only utilize ResNet-50 as our backbone, suggesting that CMANet is capable of better performance with a more powerful backbone. However, the performance of CMANet is lower than that of RDFNet on mIoU by 0.1%. Despite its slight deficiency in segmentation performance, CMANet displays an improvement in memory and computing complexity according to the model efficiency analysis. Furthermore, additional experiments are conducted to compare the utilization of HHA images and depth images. The results display that the application of HHA images can slightly improve the performance of CMANet, with a 0.3% increase on mIoU.

As shown in Table 4, we also compare the category-wise results on class IoU. CMANet performs better than RefineNet-101 and LSD-GF over 28 and 22 classes (40 classes in total), respectively. These results demonstrate the robustness and effectiveness of CMANet in indoor-scene semantic segmentation.

Due to the limited data scale of NYUDv2 dataset, we also compare the semantic-segmentation performance of CMANet on the large-scale SUN RGB-D dataset with other existing methods, following the same training and testing strategy as on the NYUDv2 dataset. The comparison results are displayed in Table 3, where CMANet achieves the best performance among all the methods on all three evaluation metrics. The semantic-segmentation performance of CMANet on the SUN RGB-D dataset further verifies its validity.

### 4.4. Ablation Study

In order to investigate the functionality of the proposed network and its processing modules, extensive ablation experiments are performed on the NYUDv2 dataset. Each experiment is conducted with the same hyper-parameter settings during training and testing periods.

The ablation study w.r.t. CMRGs is performed to verify the functionality of CMRGs in different encoding stages. As displayed in Table 5, among the first three defective models, each of the first four defective models removes certain CMRGs, but the fifth contains them all, i.e., the original CMANet. It is interesting to find that the second defective models ($G_{3}$, $G_{4}$ and $G_{5}$ are removed) outperform the first one ($G_{1}$, $G_{2}$ are removed). Furthermore, with the gradual stacking of the CMRG from lower to higher stages, the performances of the defective models become better, with the best performance using the CMRG in all stages. From these facts, it is clear that the cross-modality fusion, i.e., the utilization of CMRGs, plays a vital role in performance improvement, especially in the earlier stages. This can be recognized by the fact that the low-level features are rough and compatible in a CNN-based model. The original CMANet performs the best, proving that the multi-stage cross-modality fusion is effective and the CMRGs are mutually reinforcing.

Additionally, we also conduct an ablation study on CMRGs, skip connections, and pyramid supervision to evaluate the effect of these strategies; the results are displayed in Table 6. The first defective model is the baseline without any strategy; the second, third, and fourth defective models remove one corresponding strategy at a time; the fifth is the original model. We compare them on mean accuracy and mean IoU, showing that the order of influence from most significant to most minor is CMRGs, skip connections, pyramid supervision. According to the results, CMRGs can effectively improve the performance of semantic segmentation, while skip connections and pyramid supervision provide slight improvements.

### 4.5. Model Efficiency Analysis

Complexity in terms of time and space is an important metric to evaluate the efficiency of the model. Accordingly, we compare our CMANet with [13,53] to verify the model’s efficiency. According to the results displayed in Table 7, our method achieves 23.8% parameters reduction and 19.4% FLOPs reduction compared to RDFNet [13] while maintaining almost the same performance. Meanwhile, CMANet outperforms 3DGNN [53] with a 20.5% inference time reduction and a 4.5% mean IoU improvement. However, CMANet has a larger number of parameters than 3DGNN and a lower mean IoU than RDFNet-50. Consequently, it can be seen that CMANet achieves a balance between model complexity and accuracy, which will be further improved in our future research.

### 4.6. Visualization

For the purpose of evaluating the performance of semantic segmentation the analysis is supposed to be not only quantitative but also qualitative; we should pay more attention to the interpretability of the results. Therefore, we conduct visualizations of the CMANet results.

**Semantic Segmentation Qualitative Visual Results** In Figure 8, we visualize some typical examples of semantic segmentation with our baseline, the defective models, and our proposed method CMANet. In Figure 8a, the bedroom has few objects on the bed, whereas Figure 8b has disorganized items on the bed. Figure 8c represents the hallway scene with a complex structure. Figure 8d has an obvious lighting imbalance in the bedroom. In Figure 8e, the table is cluttered with small and numerous objects. Figure 8f presents a scene in which there are not only strong lighting conditions, but also many overlapped and similar-texture objects. Compared to other methods, CMANet promotes semantic-segmentation results from the perspective of details and the misclassification phenomenon.

**Pyramid Supervision Visualization** We conduct visualizations of the pyramid supervision by generating semantic maps from the final output and four side outputs; meanwhile, the performance of some randomly sampled examples is illustrated in Figure 9. The fourth to eighth columns denote the final output, and the four side outputs have spatial resolutions of 1/2, 1/4, 1/8, and 1/16, respectively, the height and width of the final output. As the semantic-segmentation of the outputs progresses from right to left, the refinement of pixel classification gradually improves. However, the results demonstrate that the output with a lower spatial resolution has a more remarkable performance on large-object (wall, TV, etc.) segmentation, as well as edge extraction, owing to its larger receptive field. In this way, the supervision in low-spatial-resolution outputs assists the higher ones via the recognition of boundaries and large objects; meanwhile, the supervision in high-spatial-resolution outputs can refine the semantic information. As a result, the pyramid supervision enables the enhancement of the final result via multi-scale semantic analysis.

**Channel Attention Visualization** To verify the effectiveness of the channel-attention mechanism, which can enhance the advantages and filter the drawback, we conduct visualizations of the RGB and HHA channel-refined features. In Figure 10, we randomly select two high-weighting feature maps from the channel-refined features. As shown in Figure 10a, the RGB feature maps focus on the significant texture regions (e.g., the wall hangings, the curtains), while the HHA feature maps are mainly concerned with the significant structure regions (e.g., the office chair, the bookshelf), as illustrated in Figure 10b. The results demonstrate that the CMRG enhances feature extraction by paying attention to essential features.

**Cross-Modality Refine Gate Visualization** In order to understand refinement by the CMRG, we visualize the output of the CMRG-refined features. As illustrated in Figure 11, we conduct visualizations on some typical feature maps on two random sampled examples. In the first row of Figure 11, the CMRG refinement enhances the regions that share the same labels, such as the sofa, the floor, or the wall, while in the second row, the refinement emphasizes the table and objects on the table. As a result, the CMRG is capable of effectively building connections among the regions with similar characteristics, even if they are geographically dispersed.

## 5. Conclusions

In this paper, we proposed a novel CMANet method for indoor-scene semantic segmentation, which utilizes HHA and RGB images to enhance the robustness of segmentation in indoor scenes. According to our experiments, CMANet not only facilitates the learning process via the enhancement of the representation, robustness, and discrimination of the feature embedding, but also takes the advantage of cross-modality features. CMANet employs the encoder–decoder architecture. The encoder has a two-parallel-branch backbone that can extract and aggregate the specific features from RGB and HHA; meanwhile, the decoder generates multi-scale semantic maps that can improve the final segmentation results. Specifically, we designed the CMRG, which is the most crucial component in CMANet. The CMRG employs a sequence of cross-modality channel- and spatial-attention modules. The channel-attention module is responsible for capturing ’what’ is important, whereas the spatial-attention module is responsible for determining ’where’ should be prioritized. The CMRG filters the noisy information and integrates the features from different modalities (RGB and HHA). The CMRG can effectively enhance representations from both modalities by selecting key features and establishing connections between relevant regions. Additionally, we employ a bi-directional multi-step propagation strategy to provide assistance in propagating. The results of the ablation study and visualization demonstrate the significance of each proposed component. The experiments on the NYUDv2 and SUN RGB-D datasets verify the robustness and effectiveness of CMANet, and the results illustrate that the network outperforms the existing indoor-scene semantic-segmentation methods and achieves a new state-of-art performance. In our future research, we will focus more on increasing the efficiency of our network by reducing time and space complexity. Moreover, we will consider applying semi-supervised or weakly-supervised learning strategies for indoor-scene semantic segmentation due to the limited dataset scale and inaccurate data labeling.

## Figures and Tables

**Figure 1 sensors-22-08520-f001:**
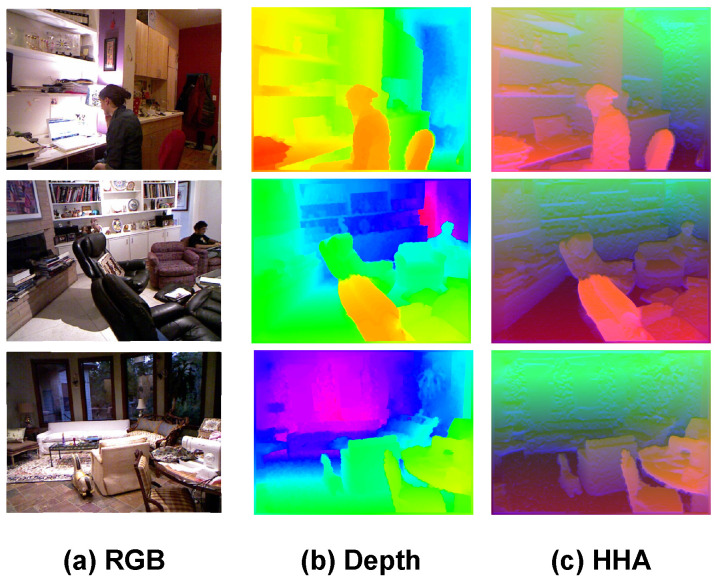
Comparisons: (**a**) RGB images, (**b**) one-channel depth images, and (**c**) three-channel HHA images.

**Figure 2 sensors-22-08520-f002:**
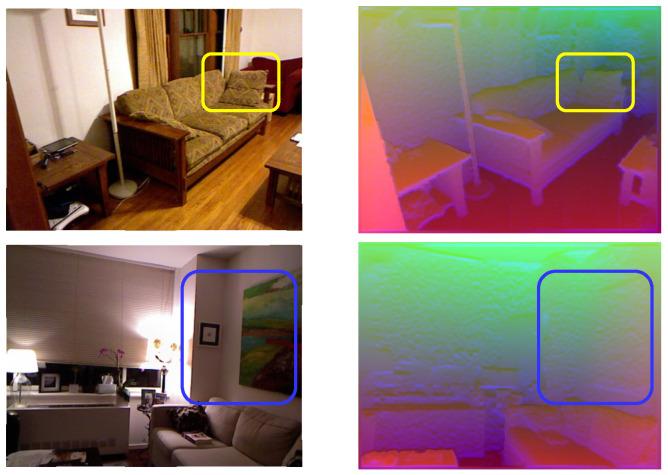
Some challenging samples of semantic-segmentation tasks. The yellow bounding box indicates the complex sample on solely RGB images, whereas the blue indicates the complex sample with solely HHA images.

**Figure 3 sensors-22-08520-f003:**
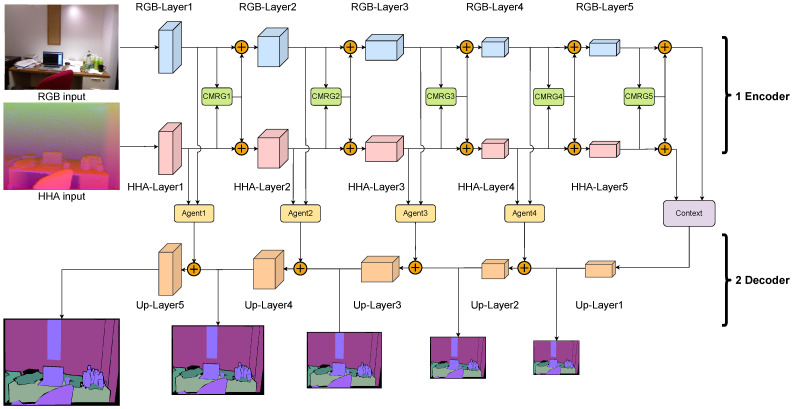
Illustration of the framework of CMANet. CMANet has the encoder–decoder architecture: (**1**) The encoder extracts RGB and HHA features with ResNet [44] backbone. (**2**) The decoder is an up-sampled backbone composed of several standard residual blocks.

**Figure 4 sensors-22-08520-f004:**
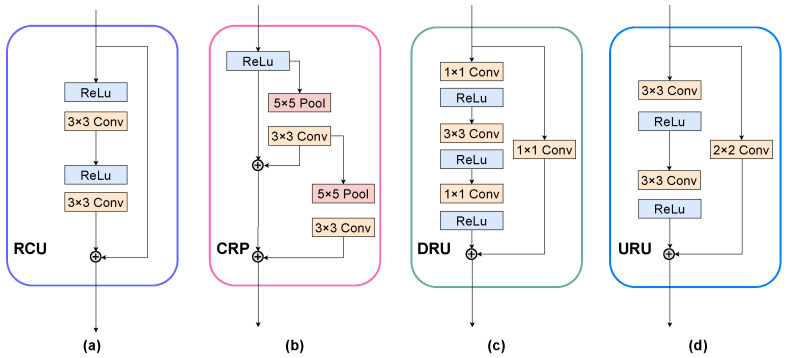
Structure of residual units: (**a**) Residual Convolutional Unit (RCU): a standard residual convolutional unit, which is an adaptive convolution that has two standard 3×3 layers with shortcut connection; (**b**) Chained Residual Pooling (CRP): a chain of pooling blocks (two blocks) with shortcut connection, each of which contains of a 5×5 max-pooling layer and a 3×3 convolution layer; (**c**) Downsample Residual Unit (DRU): a downsample residual unit in the (ResNet-50) backbone; (**d**) Upsample Residual Unit (URU): an upsample residual unit that we propose in the decoder.

**Figure 5 sensors-22-08520-f005:**
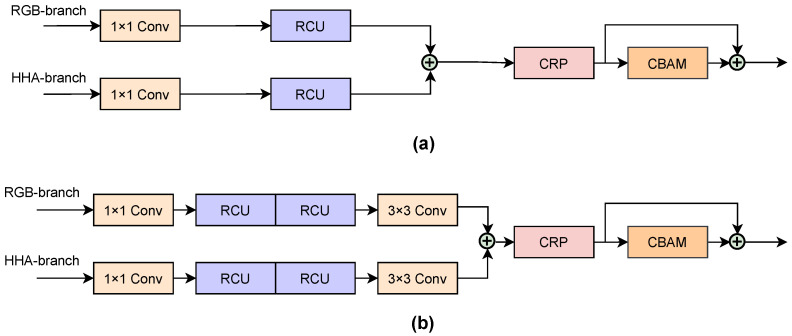
Structures of processing modules: (**a**) the structure of the Agent module; (**b**) the structure of the Context module.

**Figure 6 sensors-22-08520-f006:**
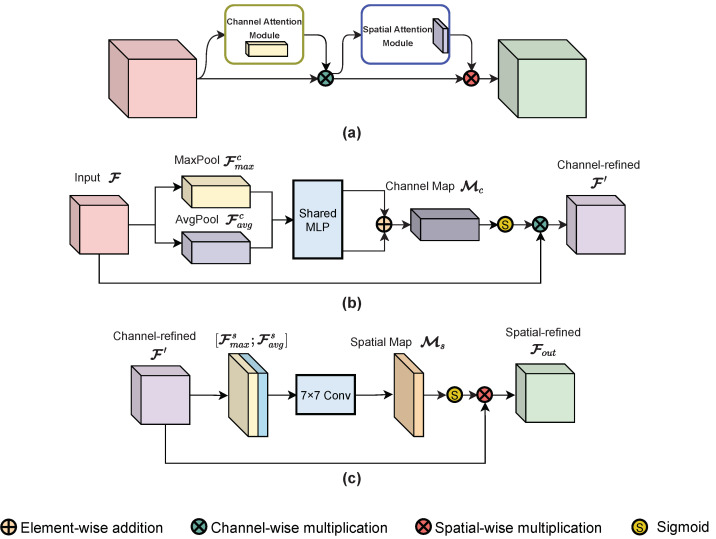
Structures of CBAM and its sub-modules: (**a**) the structure of CBAM, which is comprised of a channel-attention module and a spatial-attention module in sequence; (**b**) the structure of the channel-attention module; (**c**) the structure of the spatial-attention module.

**Figure 7 sensors-22-08520-f007:**
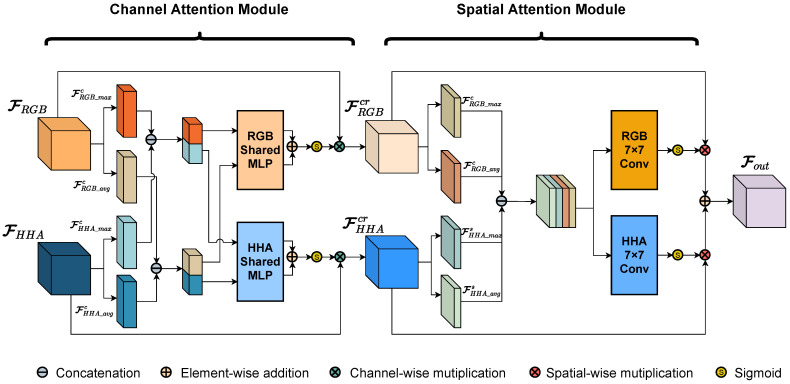
Structure of the Cross-Modality Refine Gate (CMRG).

**Figure 8 sensors-22-08520-f008:**
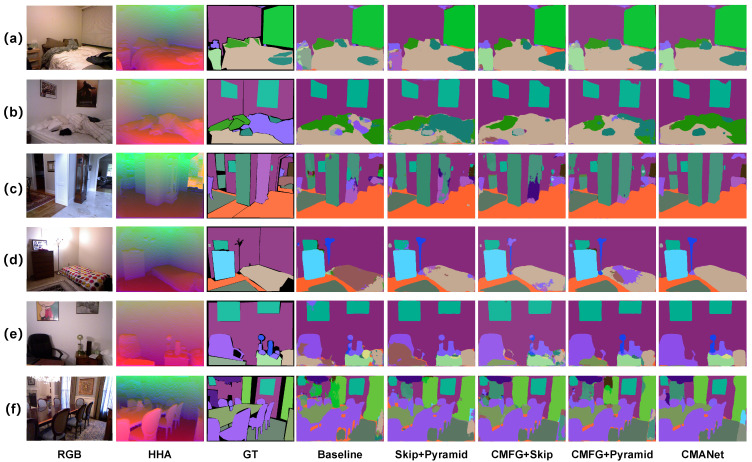
Semantic segmentation qualitative visual results on NYUDv2 dataset. The (**a**) represents the bedroom has few objects on the bed; (**b**) has disorganized items on the bed; (**c**) represents the hallway scene with a complex structure; (**d**) has an obvious lighting imbalance in the bedroom; (**e**) represents the table which is cluttered with small and numerous objects; (**f**) presents a scene in which there are not only strong lighting conditions, but also many overlapped and similar-texture objects.

**Figure 9 sensors-22-08520-f009:**
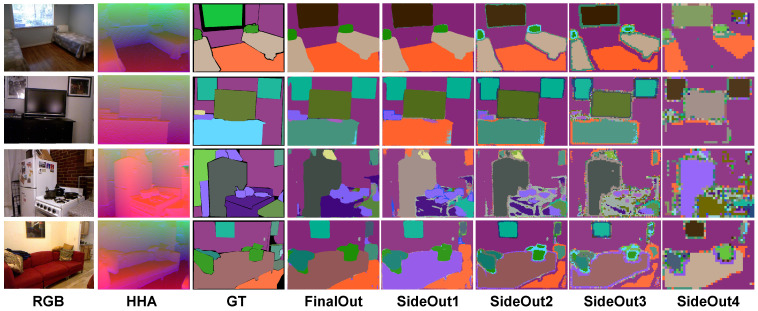
Visualization of pyramid supervision.

**Figure 10 sensors-22-08520-f010:**
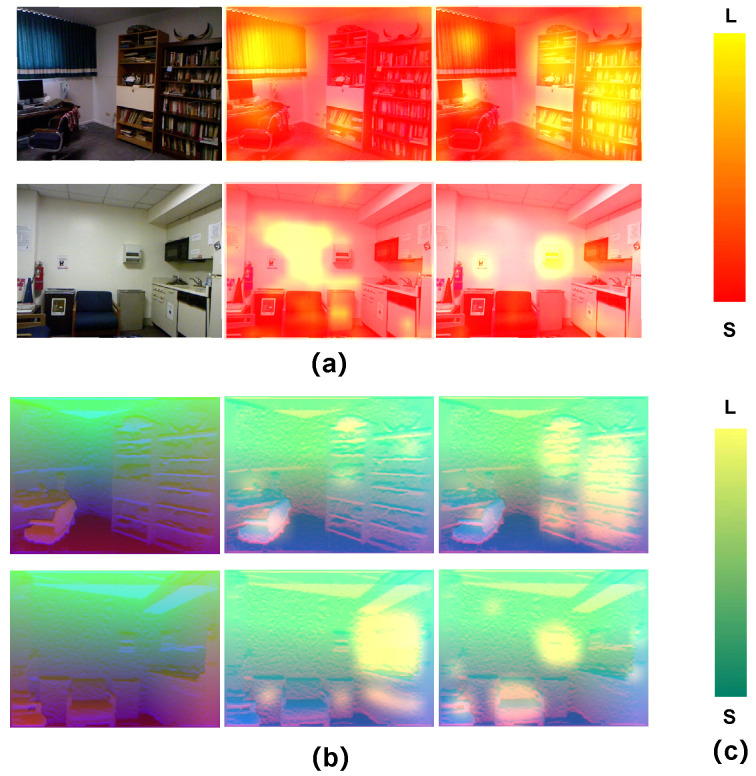
Visualization of channel attention. (**a**) The visual results on RGB channel-refined features; (**b**) the visual results on HHA channel-refined features; (**c**) the color bar.

**Figure 11 sensors-22-08520-f011:**
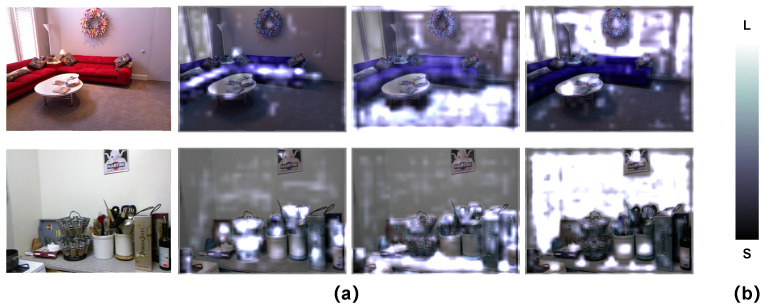
Visualization of the cross-modality refine gate. (**a**) Visual results of the output of the CMRG; (**b**) the color bar.

**Table 1 sensors-22-08520-t001:** Encoder (ResNet-50) and decoder configuration.

Block	Encoder			Block	Decoder		
	Input	Output	Units		Input	Output	Units
Layer1	3	64	-	Trans1	512	256	6
Layer2	64	256	3	Trans2	256	128	4
Layer3	256	512	4	Trans3	128	64	3
Layer4	512	1024	6	Trans4	64	64	3
Layer5	1024	2048	3	Trans5	64	64	3

**Table 2 sensors-22-08520-t002:** Comparison of CMANet with state-of-the-arts methods on the NYUDv2 dataset in 40 classes. The results are reported in terms of percentage (%) of pixel accuracy, mean accuracy, and mean IoU. The results of other methods originate from the corresponding citation.

Method	Data	Pixel Acc. (%)	Mean Acc. (%)	Mean IoU. (%)
Gupta et al. [46]	RGB-D	60.3	35.1	28.6
Deng et al. [41]	RGB-D	63.8	-	31.5
FCN-16s [4]	RGB-HHA	65.4	46.1	34.0
Wang et al. [14]	RGB-D	-	47.3	-
Context [42]	RGB	70.0	53.6	40.6
STD2P [52]	RGB-D	70.1	53.8	40.1
3DGNN [53]	RGB-HHA	-	55.7	43.1
Depth-aware [54]	RGB-HHA	-	56.3	43.9
LSD-GF [51]	RGB-HHA	71.9	60.7	45.9
RefineNet-101 [45]	RGB	72.8	57.8	44.9
RDFNet-50 [13]	RGB-HHA	74.8	60.4	47.7
RAFNet-50 [43]	RGB-D	73.8	60.3	47.5
CMANet-50-depth (ours)	RGB-D	73.9	59.8	47.3
CMANet-50 (ours)	RGB-HHA	74.2	60.2	47.6

**Table 3 sensors-22-08520-t003:** Comparison of CMANet with state-of-the-art methods on the SUN RGB-D dataset in 37 classes. The results are reported in terms of percentage (%) of pixel accuracy, mean accuracy, and mean IoU.

Method	Data	Pixel Acc. (%)	Mean Acc. (%)	Mean IoU. (%)
SegNet [5]	RGB	71.2	45.9	30.4
FuseNet [19]	RGB-D	76.3	48.3	37.3
Depth-aware [54]	RGB-HHA	-	53.5	42.0
Context [42]	RGB	78.4	53.4	42.3
3DGNN [53]	RGB-D	-	57.0	45.9
RefineNet-101 [45]	RGB	80.4	57.8	45.7
RedNet-34 [18]	RGB-D	80.8	58.3	46.8
CMANet-50 (ours)	RGB-HHA	81.1	59.3	47.2

**Table 4 sensors-22-08520-t004:** Comparison of CMANet with state-of-the-art methods on the NYUDv2 dataset in 40 classes. The results are reported in terms of percentage (%) of IoU.

Method	Wall	Floor	Cabinet	Bed	Chair	Sofa	Table	Door	Window	Bookshelf
Gupta et al. [46]	68.0	81.3	44.9	65.0	47.9	29.9	20.3	32.6	9.0	18.1
Deng et al. [41]	65.6	79.2	51.9	66.7	41.0	55.7	36.5	20.3	33.2	32.6
FCN-16s [4]	69.9	79.4	50.3	66.0	47.5	53.2	32.8	22.1	39.0	36.1
STD2P [52]	72.7	85.7	55.4	73.6	58.5	60.1	42.7	30.2	42.1	41.9
LSD-GF [51]	78.5	87.1	56.6	70.1	65.2	63.9	46.9	35.9	47.1	48.9
RefineNet-101 [45]	77.5	82.9	58.7	65.7	59.1	57.8	40.1	36.7	45.8	42.8
CMANet-50 (ours)	77.7	86.2	59.6	72.5	60.3	61.1	43.3	35.5	43.8	38.6
**Method**	**Picture**	**Counter**	**Blind**	**Desk**	**Shelf**	**Curtain**	**Dresser**	**Pillow**	**Mirror**	**Mat**
Gupta et al. [46]	40.3	51.3	42.0	11.3	3.5	29.1	34.8	34.4	16.4	28.0
Deng et al. [41]	44.6	53.6	49.1	10.8	9.1	47.6	27.6	42.5	30.2	32.7
FCN-16s [4]	50.5	54.2	45.8	11.9	8.6	32.5	31.0	37.5	22.4	13.6
STD2P [52]	52.9	59.7	46.7	13.5	9.4	40.7	44.1	42.0	34.5	35.6
LSD-GF [51]	54.3	66.3	51.7	20.6	13.7	49.8	43.2	50.4	48.5	32.2
RefineNet-101 [45]	60.1	56.8	61.4	22.6	12.3	53.5	38.3	39.6	38.7	29.7
CMANet-50 (ours)	60.9	62.5	56.1	21.7	10.0	56.1	50.1	46.4	45.8	37.2
**Method**	**Cloths**	**Ceiling**	**Books**	**Refridg**	**TV**	**Paper**	**Towel**	**Shower**	**Box**	**Board**
Gupta et al. [46]	4.7	60.5	6.4	14.5	31.0	14.3	16.3	4.2	2.1	14.2
Deng et al. [41]	12.6	56.7	8.9	21.6	19.2	28.0	28.6	22.9	1.6	1.0
FCN-16s [4]	18.3	59.1	27.3	27.0	41.9	15.9	26.1	14.1	6.5	12.9
STD2P [52]	22.2	55.9	29.8	41.7	52.5	21.1	34.4	15.5	7.8	29.2
LSD-GF [51]	24.7	62.0	34.2	45.3	53.4	27.7	42.6	23.9	11.2	58.8
RefineNet-101 [45]	24.4	66.0	33.0	52.4	52.6	31.3	36.8	23.6	11.1	63.7
CMANet-50 (ours)	21.1	75.3	33.1	55.1	63.3	30.1	40.1	32.1	14.3	62.5
**Method**	**Person**	**Stand**	**Toilet**	**Sink**	**Lamp**	**Bathtub**	**Bag**	**Othstr**	**Othfurn**	**Otherprop**
Gupta et al. [46]	0.2	27.2	55.1	37.5	34.8	38.2	0.2	7.1	6.1	23.1
Deng et al. [41]	9.6	30.6	48.4	41.8	28.1	27.6	0.0	9.8	7.6	24.5
FCN-16s [4]	57.6	30.1	61.3	44.8	32.1	39.2	4.8	15.2	7.7	30.0
STD2P [52]	60.7	42.2	62.7	47.4	38.6	28.5	7.3	18.8	15.1	31.4
LSD-GF [51]	53.2	54.1	80.4	59.2	45.5	52.6	15.9	12.7	16.4	29.3
RefineNet-101 [45]	78.6	38.6	68.4	53.2	45.9	32.9	14.6	32.9	18.7	36.4
CMANet-50 (ours)	77.3	40.8	70.9	58.9	47.9	57.3	13.6	31.2	19.1	38.1

**Table 5 sensors-22-08520-t005:** Ablation study of CMRGs on NYUDv2 dataset in 40 classes. The results are reported in terms of percentage (%) of mean IoU. *G* denotes the CMRG in different encoding stage. The best performance is marked in bold.

Backbone	$G_{1}$	$G_{2}$	$G_{3}$	$G_{4}$	$G_{5}$	Mean IoU. (%)
ResNet-50			✓	✓	✓	46.5
ResNet-50	✓	✓				46.7
ResNet-50	✓	✓	✓			47.2
ResNet-50	✓	✓	✓	✓		46.8
ResNet-50	✓	✓	✓	✓	✓	**47.6**

**Table 6 sensors-22-08520-t006:** Ablation study of CMRGs, skip connections, and pyramid supervision on NYUDv2 dataset in 40 classes. The results are reported in terms of percentage (%) of mean IoU. The best performance is marked in bold.

Backbone	Baseline	CMRG	Skip	Pyramid	Mean IoU. (%)
ResNet-50	✓				44.2
ResNet-50	✓	✓		✓	46.8
ResNet-50	✓		✓	✓	45.9
ResNet-50	✓	✓	✓		47.0
ResNet-50	✓	✓	✓	✓	**47.6**

**Table 7 sensors-22-08520-t007:** Comparison of model complexity. The results are reported in terms of the number of parameters (million), the computing complexity of FLOPs (gigabyte), inference time (millisecond), and mean IoU (%). The inference time and FLOPs are evaluated on an Nvidia 1660Ti GPU with the RGB input of 3×480×640 and the HHA input of 3×480×640.

Method	Backbone	Params. (M)	FLOPs. (G)	Inference Time. (ms)	Mean IoU. (%)
3DGNN [53]	ResNet-101	47.3	-	492.5	43.1
RedNet-101 [18]	ResNet-101	121.2	-	268.5	-
RefineNet-101 [45]	ResNet-101	126.0	-	248.4	44.9
RDFNet-50 [13]	ResNet-50	153.3	168.9	368.2	47.7
CMANet-50 (ours)	ResNet-50	117.8	137.2	391.5	47.6

## Data Availability

The data presented in this study are available on request from the corresponding author.

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
