# Peer review of "CMANet: Cross-Modality Attention Network for Indoor-Scene Semantic Segmentation"

_sensors, 2022, doi:10.3390/s22218520_

Round 1

Reviewer 1 Report

This paper proposes a cross-modality attention network for indoor scene semantic segmentation problem. In general, the idea is interesting, and here are some comments:

1.Appropriately expand the understanding of the research direction of abstracts and keywords.

2.The innovation points of CMANet can be explained more, and please add more application backgrounds.

3.CMRG can be compared with CBAM in more detail.

4.The data enhancement strategy can describe the results in more detail

5.Opinions should be put forward for future work

6.The consecutive short sentences in the abstract are not refined enough

7. The introduction of the conclusion is too short, and some schematic diagrams can be added for explanation.

8. Other segmentation methods are suggested to added in the introduction section, such as active contour model, e.g., An active contour model driven by adaptive local pre-fitting energy function based on Jeffreys divergence for image segmentation, Expert Systems with Applications; A level set method based on additive bias correction for image segmentation, Expert Systems with Applications; A hybrid active contour model based on pre-fitting energy and adaptive functions for fast image segmentation, Pattern Recognition Letters

Reviewer 2 Report

This work proposes a so-called Cross-Modality Attention Network (CMANet), that deals with the complexity of extracting and integrating multi-modality features, in this case, RGB and HHA (3 channel images resulting from applying the HHA method to RGB-D sensor information), for semantic segmentation.

While HHA images are able to encode geometric information it is sometimes difficult to distinguish, for example, objects with different textures but in the same geometric plane. The authors argue that this can be solved using a combination of both RGB and HHA. CMANet is an encoder-decoder architecture using a ResNet and an upsample ResNet. To deal with multi-modality the encoder consists of two branches (one for each modality) that gradually downsample the features while sharing features through a Structure of Cross-Modality Refine Gate an attention-based module based on CBAM and designed for cross-modality.

The paper introduces the problem well and has an adequate description of related work. However, it doesn’t properly cite work that introduces chained residual pooling, downsample residual unit, and upsample residual unit.

The network and the new modules used in the approach are well explained and the text is clear.

However, the results raise some questions. While the improvement is interesting the methods that are used for comparison are more than three years old. More recent methods, that the authors cite but don’t compare with (in Table 4), like RDFNet-101 or even RAFNet-50, are similar approaches and one of them even uses the same backbone, with similar results. Furthermore, RAFNet-50 uses RGB-D, which raises some questions about the real value of the HHA in comparison to a depth map.

Even though, it is interesting to see what the network is capable of even using a smaller backbone. it would be interesting to see the gains from using larger network, such as a ResNet 101.

Reviewer 3 Report

Please make the comparisons with the existent works that listed in Table-3, by the follow factors

1. computational complexity and

2. execution time

3. memory usage 

Round 2

Reviewer 1 Report

The authors have addressed all my comments.

Author Response

Dear Reviewers,

We would like to take this opportunity to thank you again for all your time involved and this great opportunity for us to improve the manuscript. We hope you will find this revised version satisfactory.

Sincerely,

The Authors

Reviewer 2 Report

The authors did address all major comments except one, and the paper is almost ready for acceptance.

However, I feel that that missing point is an important one and needs to be corrected. In reply to question 2, the authors state that the comparison with RDFNet is included in table 7. However, the request was to include the accuracy comparison with RDFNet in table 2, which is the relevant table for this purpose. 

I therefore think that the inclusion of the performance of RDFNet in table 2 (and not only in table 7) is critical and needs to be done before the paper could be accepted. Without it, I cannot state that the paper presents a fair comparison with the state of the art.
